# Clinical Applications of Endobronchial Ultrasound (EBUS) Scope: Challenges and Opportunities

**DOI:** 10.3390/diagnostics13152565

**Published:** 2023-08-01

**Authors:** Davide Biondini, Mariaenrica Tinè, Umberto Semenzato, Matteo Daverio, Francesca Scalvenzi, Erica Bazzan, Graziella Turato, Marco Damin, Paolo Spagnolo

**Affiliations:** 1Respiratory Disease Unit, Department of Cardiac, Thoracic, Vascular Sciences and Public Health, University of Padova, 35128 Padova, Italy; 2Department of Medicine, University of Padova, 35128 Padova, Italy

**Keywords:** endobronchial ultrasound, transbronchial needle aspiration, bronchoscopy

## Abstract

Endobronchial Ultrasound (EBUS) has been widely used to stage lung tumors and to diagnose mediastinal diseases. In the last decade, this procedure has evolved in several technical aspects, with new tools available to optimize tissue sampling and to increase its diagnostic yield, like elastography, different types of needles and, most recently, miniforceps and cryobiopsy. Accordingly, the indications for the use of the EBUS scope into the airways to perform the Endobronchial Ultrasound–TransBronchial Needle Aspiration (EBUS-TBNA) has also extended beyond the endobronchial and thoracic boundaries to sample lesions from the liver, left adrenal gland and retroperitoneal lymph nodes via the gastroesophageal tract, performing the Endoscopic UltraSound with Bronchoscope-guided Fine Needle Aspiration (EUS-B-FNA). In this review, we summarize and critically discuss the main indication for the use of the EBUS scope, even the more uncommon, to underline its utility and versatility in clinical practice.

## 1. Introduction

Transbronchial needle aspiration (TBNA) for sampling mediastinal lymph nodes was first described in 1949 by Schieppati [1], an Argentinian physician, via a rigid bronchoscope. This technique was relatively neglected until 1978 when Wang and colleagues successfully diagnosed lung carcinoma adjacent to the trachea and paratracheal nodal metastasis using an esophageal varices needle [2]; one year later, a new needle to insert in the fiberoptic bronchoscope was developed [3]. Finally, in 1983, Wang reported on 35 patients who underwent TBNA and concluded that the procedure was relatively safe and allowed performing both a diagnosis of lung cancer and disease staging [4].

Since then, TBNA has become a widely used technique to sample lesions, including nodules and masses, which are outside of but adjacent to the central airways, by inserting the needle in the site suggested by a careful analysis of the chest Computed Tomography and endobronchial anatomic landmarks [5].

In 2002, the development of the convex probe to perform endobronchial ultrasound was a turning point because it allowed the bronchoscopist to perform real-time transbronchial needle aspiration guided by endobronchial ultrasound (EBUS-TBNA) [6]. The high diagnostic yield and safety profile of EBUS-TBNA were soon recognized, making this technique very popular worldwide [7]. However, it was only after the ASTER trial [8], which showed similar sensitivity for EBUS-TBNA compared to mediastinoscopy (85% vs. 79%, respectively), with lower complications and fewer unnecessary thoracotomies associated with the former, that EBUS-TBNA became the technique of choice for staging the mediastinum in lung cancer [9,10].

More recently, the clinical applications of EBUS have expanded. Indeed, due to the larger amount of tissue required by the pathologist to perform not only a correct diagnosis of cancer but also a precise molecular analysis, the endoscopist had to upgrade his technique routine to provide sample adequacy. As such, new opportunities to use the EBUS scope outside the “conventional” endobronchial route are emerging, even beyond the thoracic boundaries. Indeed, to evaluate lesions that cannot be accessed through the airways, it can be used through the gastrointestinal tract to perform an endoscopic ultrasound with bronchoscope-guided fine needle aspiration (EUS-B-FNA).

The aim of this review is to summarize the various applications of the EBUS scope, also highlighting the most recent approaches and techniques.

## 2. EBUS-TBNA in the Molecular Profiling of Lung Cancer

In recent years, research on molecular mutations in non-small cell lung cancer (NSCLC) has increased our knowledge of biological pathways driving tumor development; the identification of mutations within the epidermal growth factor receptor (EGFR) gene is one such example [11]. Owing to the beneficial effect of molecularly targeted therapies and immune checkpoint inhibitors on both overall and progression-free survival in NSCLC, the importance of obtaining adequate specimens for precise tumor genotyping cannot be underestimated [12]. Several prospective trials have established the non-inferiority, or even superiority, of EBUS-TBNA compared to surgical mediastinoscopy to achieve a correct diagnosis and mediastinal staging [13,14,15], with an excellent safety profile and a complication rate lower than 1% [12]. However, the capability of EBUS-TBNA, which is a cytological methodology, to provide a complete molecular analysis is still debated [16,17,18]. 

Several aspects of the EBUS-TBNA technique may impact the adequacy of the sample for molecular analysis, such as the number of needle passes, the use of rapid on-site evaluation or the suction during the needle aspiration and the choice of specimen preservation.

Lee et al. reported that three needle passes for each target site are sufficient to secure a diagnosis of NSCLC [19]; however, to obtain an exhaustive molecular analysis, it is recommended that supplementary needle passes are performed [20,21,22]. This was confirmed by a large cohort study and a recent meta-analysis on 21 studies (n = 1.175 patients) in which more passes are reported to increase the sensitivity for EGFR mutation and anaplastic lymphoma kinase (ALK) rearrangements and to provide adequate samples for next-generation sequencing (NGS) [23,24].

Rapid on-site evaluation (ROSE), which provides real-time pathologic evaluation of sample adequacy, was initially developed to enhance the diagnostic yield of conventional TBNA [25,26,27], but its role in EBUS-TBNA is still controversial. Indeed, some studies have shown that ROSE does not increase the diagnostic yield of EBUS-TBNA [28] but reduces the number of punctures per site (2.2 vs. 3.1, respectively) and the need for further diagnostic procedures (11% vs. 57%, respectively) [29]. Moreover, in the context of molecular profiling, Trisolini and colleagues reported that performing ROSE during EBUS-TBNA increased the rate of complete tumor genotyping by up to 10% [30]. Even though this data was clinically meaningful, it did not reach statistical significance. 

Further dissecting factors that may potentially influence the performance of EBUS-TBNA, there is the utilization of high-pressure suction. This technique, compared to standard vacuum syringe suction, did not impact the diagnostic yield; however, Tsakinis reported that EBUS-TBNA with high-pressure suction was associated with bigger samples (11.2 vs. 9.1 mm^3^, *p* = 0.036) and fewer additional procedures (3.8% vs. 17.5%, *p* = 0.042) to reach complete molecular profiling in NSCLC with a necrotic component [31]. 

Conversely, the complete or partial retraction of the stylet, as retrospectively assessed in 50 patients, does not seem to influence the adequacy of molecular testing [32]. 

Another variable that influences the adequacy of the sample is related to the pathologist’s choice of specimen preservation. In accordance with the College of American Pathologists guidelines, cell blocks are recommended over smear preparations because of their ability to correlate with malignant cell content and the possible conservation of more material for additional studies [33]. Gross pathological specimens can be preserved in paraffin blocks, the so-called “cell block” that can be cut and stained for histopathological analysis. This technique preserves the cytological architecture and allows for immunochemical staining, thereby providing an accurate characterization of the cancer [34]. As compared with the conventional smear method, cell block analysis can improve the yield of EBUS-TBNA by 7% and can provide data for genetic analysis in patients with adenocarcinoma [35].

In advanced lung cancer, complete molecular profiling is mandatory to initiate an appropriate treatment. Labarca et al. recently reported in a systematic review and meta-analysis that the adequacy of EBUS-TBNA to evaluate EGFR and ALK mutations was feasible in 94.5% (28 studies, n = 2497 patients) and 95% of the patients (12 studies, n = 607), respectively [36]. On the same line, Martin-Delon et al. prospectively assessed stage III and IV NSCLC to evaluate the adequacy of EBUS-TBNA samples for molecular characterization, as assessed by NGS, nCounter and immunohistochemistry (PD-L1) and to compare its performance to the reference standard of biopsy samples. EBUS-TBNA showed a high concordance with biopsies: 100% for NGS and nCounter and 88.9% for PD-L1 [37]. In the same study, ROSE was routinely performed, and the average number of needle passes was six, much higher than suggested [20], possibly favoring the good result of EBUS-TBNA sampling. Indeed, 92.8% of TBNA (compared to 75.8% obtained by bronchial biopsy) provided adequate material to successfully perform the analysis of all tumor predictors. Although PD-L1 testing was adequate with the majority of EBUS-TBNA samples, a low agreement with biopsies was observed in PD-L1 immunohistochemistry scores when considering the three categories (negative, low- and high-positive) [37]. Indeed, as previously mentioned, PD-L1 testing on EBUS-TBNA or other tissue specimens may provide differing results, as shown by the concordance rates ranging between 69.8% and 91.3% reported in the literature [38,39,40]. A possible reason for such a discrepancy is related to the intrinsic spatial and temporal heterogeneity of PD-L1 expression that characterizes NSCLC [41], which increases the risk of false negatives in small samples with low tumor cellularity, such as those obtained by EBUS-TBNA. The potential misclassification of PD-L1 status in EBUS-TBNA specimens remains a considerable risk, especially when a cutoff of PD-L1 tumor proportion score ≥ 50% is applied. Nonetheless, a recent real-life study suggests that EBUS-TBNA results are as valuable as histological specimens to guide immune checkpoint inhibitor therapy, with comparable treatment outcomes, supporting the regular testing for PD-L1 in EBUS-TBNA samples [42].

Notably, an innovative technique to provide additional DNA for NGS analysis is the isolation of free-floating DNA, which is visible after centrifugation of the needle aspiration supernatant, thus reducing the need for repeated biopsy and turnaround time [43,44,45]. However, the evaluation of supernatant needs to be further studied in larger cohorts of patients to establish its diagnostic accuracy and impact on NGS analysis [28].

Regarding the early stages of NSCLC, traditionally, the treatment refers to surgical resection, less frequently associated with radio-chemotherapy or neoadjuvant/adjuvant chemotherapy [46]. In the latter scenario, favorable results have recently emerged. Adjuvant therapy in EGFR mutated NSCLC, as well as preoperative immunotherapy in early stages, was associated with fewer relapses or post-operative tumor progression [47,48]. In the ADAURA study, osimertinib, a third-generation tyrosine kinase inhibitor that was administered as an adjuvant in EGFR mutated NSCLC, was associated with an 80% reduction of recurrence compared to placebo after surgical resection of the tumor. The NeoADAURA study, a clinical trial of osimertinib in a neoadjuvant setting in early operable stages of NSCLC, is ongoing (NCT04351555) [49]. Preoperative molecular characterization of lung cancer is critical, as the evaluation of EGFR and ALK status is required before considering neoadjuvant immunotherapy or neoadjuvant immunotherapy and chemotherapy [50]. However, because the small size of samples obtained by bronchoscopy and EBUS-TBNA may represent a challenge for pathologists, the optimization of the sample for analysis will require teamwork between pulmonologists, oncologists and pathologists [51,52].

In view of these limitations, current guidelines do not recommend molecular testing in stages I–III with the broad molecular assessment suggested only in metastatic NSCLC [53]. Nonetheless, recently updated guidelines, including from the National Comprehensive Cancer Network^®^ (NCCN^®^), Canadian, and Chinese/Asian guidelines, suggest testing at least EGFR mutations [53]. 

Collectively, accumulating evidence highlights the need to optimize the management of diagnostic samples based on lung cancer stage and the importance of early definition of molecular/immune prognostic markers not only in metastatic disease but also in the pre-operative stage of resectable cancer.

## 3. Elastography

There are several echographic features during EBUS-TBNA that may raise the suspicion of a malignant lymph node, such as echogenicity, margins, diameter, or the presence of necrosis. However, none of these characteristics has consistently been associated with malignancy [54]. 

In this regard, endobronchial elastography is a relatively new technique combined with the endoscopic ultrasound procedure that has the potential to differentiate malignant versus benign lymph nodes. Elastography assesses the relative elasticity or stiffness of the tissue according to the degree of deformation after compression in a color-based output. A blue signal represents hard tissue, such as malignancy, while benign lesions, which are softer, exhibit a red-green color. 

Tissue elasticity can be measured by a qualitative or semi-quantitative modality.

The most widely used qualitative classification is the Izumo score, which identifies three types of findings: predominantly nonblue (type 1), partially blue (type 2) and predominantly blue (type 3) [55]. However, even though type 3 lymph nodes suggest malignant involvement, up to 20% may have pathological discordance [56,57]. Moreover, no correlation with the maximum standardized uptake value of positron emission tomography-computed tomography (PET-CT) images has been observed [56].

Discrepancies may arise due to different interpretations of elastography images, which is an operator-dependent evaluation. As a result, some changes have been applied to the observation method of elastography, like the stiffness ratio, strain histogram or strain ratio, to predict the elasticity of the lesion [58,59]. However, these semi-quantitative measurements require specific software for image processing, calculations and analysis, thus increasing the complexity of the procedure. 

Multiple factors may influence the results obtained by EBUS elastography. False positive results are mostly due to increased stiffness in cases of tuberculosis, pneumoconiosis, or sarcoidosis, while necrosis, hemorrhage or liquefaction typical of malignant lesions can induce false negatives [60,61]. 

Elastography is useful for the evaluation of multistation lymph nodes involvement and also when multiple nodes are present in the same station; indeed, it may prevent unnecessary punctures focusing on type 3 lymph nodes and avoid the spread of metastatic tissue in the same stage lymph node when performing puncture from type 1 to 3 lymph nodes. Finally, and most importantly, elastography cannot replace lymph node aspiration.

## 4. EUS-B Fine Needle Aspiration (EUS-B-FNA)

Endoscopic ultrasound (EUS) of the gastrointestinal tract is performed with a gastrointestinal echoendoscope by the gastroenterologists but also with the EBUS scope by the pulmonologists (EUS-B) to obtain a fine needle aspiration (FNA). This technique is proceeding forward because of its usefulness to sample tissue, which is not reachable with the endobronchial approach to achieve lung cancer diagnosis and a more accurate staging [62,63,64]. 

Paraesophageal pulmonary lesions could be safely and accurately sampled by EUS-B-FNA, with high diagnostic accuracy also for molecular profiling [65,66].

Regarding the staging of lung cancer, EUS-B fine needle aspiration (FNA) permits lymph node stations of 2 L, 4 L and 7 to be reached similarly to the endobronchial approach. Stations 2 R and 4 R are difficult to reach with EUS-B-FNA because usually, there is the interposition of the trachea, but they could be sampled, especially when lymph nodes are enlarged. Regarding para-aortic stations 5 and 6, they can be visualized, but the sampling is troublesome because of the interposition of great vessels. However, the transvascular approach is described (see Section 5.2), as well as station 6 sampling when it is located in the aorto-pulmonary bifurcation [67].

Notably, lymph nodes stations 8 and 9, as well as lesions below the diaphragm, are only reachable with this technique.

In fact, the European guidelines for combined EBUS and EUS recommend both techniques, endobronchial and esophageal, which can be performed in the same session [68], to stage mediastinal lymph nodes to increase the diagnostic yield compared to one technique alone [10]. Moreover, the advantage of the esophageal approach is the feasibility of performing an awake procedure in those patients with a high risk for general anesthesia or conscious sedation with a good tolerance [69].

Even though for the EBUS-TBNA, the European Respiratory Society (ERS) has launched a structured, evidence-based education program based on training in simulators with a final validated test [70], EUS-B-FNA is not yet available. However, experts in the fields and educators are developing a combined training curriculum in EUS-B-FNA to fill this gap [71]; it is reported to be easily acknowledged by an experienced bronchoscopist [72].

Indeed, in order to perform a more accurate staging of lung cancer, EUS-B-FNA is recommended for the evaluation of the left adrenal gland (LAG) for suspected metastasis [10]. Interestingly, up to 7% of patients with resectable cancer have adrenal masses, which in approximately two-thirds of cases are benign adenomas [73,74,75]. As a result, pathological assessment of suspicious adrenal glands is mandatory. Crombag and Annema reported that the diagnostic yield of LAG EUS-B-FNA was comparable to that of conventional EUS-FNA performed with gastrointestinal echoendoscope [76]. More recently, in a large cohort of patients (n = 135), the safety and feasibility of this procedure were confirmed, with nearly 90% of samples being adequate; notably, and as previously reported, one-third of the cases were neoplastic [77]. 

Although the use of EUS-B-FNA has also been reported for sampling suspected left liver lobe lesions or retroperitoneal lymph nodes, current guidelines do not make recommendations in this regard. Nevertheless, the diagnostic yield reported by Christiansen et al. was higher than 80% for left liver lobe lesions (n = 23) and 63% for retroperitoneal lymph nodes (n = 19), with no complication observed [77].

## 5. “Uncommon” Use of the EBUS Scope

### 5.1. Mediastinal Vessels Evaluation

EBUS allows for excellent visualization of the structures surrounding the airways, including pulmonary arteries. Casoni and co-workers originally reported on a case wherein pulmonary angiography did not allow to distinguish between pulmonary emboli and sarcoma [78]. EBUS showed “in real-time, a blood clot floating into the right main pulmonary artery not infiltrating the wall of blood vessel”, thus permitting a correct diagnosis of pulmonary embolism (PE) [78]. This was the first report on the use of EBUS to evaluate pulmonary emboli. Subsequently, a multicenter pilot study [79] showed that EBUS had an accuracy of 96% in detecting PE, which reached 100% if only centrally located emboli were considered.

The preferred technique to assess the severity of PE is computed tomography (CT) angiography; however, EBUS should be considered in the diagnostic flow-chart in patients in whom angiography is contraindicated, such as those with chronic renal failure, previous anaphylactic reactions to intravenous contrast agents and pregnant women [80]. Moreover, when PE is suspected in intensive care and patients cannot undergo a CT scan, EBUS can be performed bedside [81]. Indeed, the association of bedside EBUS and echocardiography allowed the diagnosis of PE as the cause of cardiac arrest in a 64-year-old woman [81].

A clinical trial currently evaluating the added value of EBUS in diagnosing PE in critically ill patients (Pilot Study to Evaluate the Role of EBUS in the Diagnosis of Acute PE in Critically Ill Patients-VEBUS-NCT04047784).

Even though most of the patients that perform EBUS already have a contrast chest CT scan available that evaluates the presence of pulmonary embolism, we believe that the endosonographic study of the central vessel during the procedure should be performed. Indeed, it does not require additional timing, as it could be done during the collection of the specimen, and it could be helpful in identifying unknown emboli, especially in high-risk patients with cancer and in those with a dated chest CT scan.

In the experience of our Interventional Pneumology Centre, the most representative case of pulmonary embolism evaluated by EBUS is shown in Figure 1.

In the differential diagnosis of pulmonary embolism, it should also be considered a neoplastic intravascular involvement. As an example, it is reported a case of primary pulmonary artery sarcoma diagnosed with EBUS-TBNA without complication in a patient non-eligible for surgery [82]. 

### 5.2. Transvascular Needle Aspiration (TVNA)

Some mediastinal stations or pathological tissue could be difficult to reach because of the interposition of a great vessel, for example, a branch of the pulmonary artery. Even though there is evidence of life-threatening bleeding after accidental puncture of the pulmonary artery during EBUS-TBNA, there are case series that confirm the relative safety of transvascular needle aspiration performed either by EBUS-TBNS or EUS-B-FNA [83]. Of the studies in the literature (n = 10, patients = 148), only three mild complications were reported (respiratory failure, moderate bleeding and atrial fibrillation), while no significant complications or major bleeding. The diagnostic yield was relatively high, with the majority of the studies higher than 90% [83]. 

However, this approach should be limited only if a clear window to the biopsy is not available and should be performed only as a last resort after all other options are excluded. Moreover, TVNA should be performed only in high-volume centers by skilled interventional pulmonologists, and all adequate precautions should be adopted as a cardiothoracic surgery on stand-by.

### 5.3. Pleural and Cardiac Lesions

In cases of suspected neoplastic pleural disease, especially when pleural effusion is present, thoracoscopy with pleural biopsy is the preferred method for diagnostic tissue sampling. However, thoracoscopy is invasive and may pose relevant risks in some patients, especially those with multiple and severe comorbidities [84]. In this regard, endoscopic ultrasound can offer a minimally invasive option for sampling pleural tissue next to large airways or esophagus. Kassier et al. sampled both mediastinal lymph nodes and a pleural mass by using a transesophageal approach during the same procedure in a case of metastatic renal cell carcinoma [85].

EBUS could be particularly useful when pleural abnormalities are not in the context of pleural effusion. Lococo et al. first sampled a single PET-positive pleural lesion located in the right costovertebral recess adjacent to the carina, suspicious of the epithelioid tumor. A final diagnosis of epithelioid-type malignant mesothelioma was subsequently confirmed following surgical resection of the lesion [86].

Therefore, in a subset of patients, EBUS-TBNA may represent a less invasive alternative to thoracoscopy for pleural sampling.

The approach to pleural lesions suspected for mesothelioma could also be performed by the esophagus with the EUS-B-FNA technique, as it was reported in four case reports [87,88,89,90], where the pleural thickness was in close proximity to the esophagus with a high diagnostic yield.

Similarly, EBUS-TBNA and EUS-B-FNA could be both used to diagnose metastatic involvement of the pleura, as reported in various case series [85,88,89] by renal cell carcinoma, adenoid cystic adenocarcinoma and Hodgkin’s B cell lymphoma.

Regarding cardiac tumors, which are extremely rare, they are usually sampled by open-heart surgery, with related mortality and morbidity. However, in patients with poor clinical conditions, endosonography could represent a minimally invasive technique to perform a diagnosis, seeing that it provides an excellent view of the left atrium. Two cases of EUS-B-FNA of left atrial masses have been reported with high accuracy and no adverse events [91].

Notably, this technique should be performed by experienced operators after having excluded alternative diagnostic approaches.

### 5.4. Pleural/Pericardial Effusion and Ascites

Endoscopic ultrasound (EUS) can offer an alternative guide when the pleural effusion is loculated or is located toward the mediastinum, and standard thoracentesis cannot be performed. Endoscopic US (EUS) guidance to drain pleural effusion was initially reported in a retrospective study in 2008 by DeWitt and colleagues. [92]. In nine cases, the EBUS scope was introduced in the esophagus and was successfully applied to drain pleural effusion, thus providing adequate samples for cytological analysis without significant complications. Similarly, Cocciardi et al. drained a loculated pleural effusion not accessible through traditional thoracentesis [93].

The risk of infection should be considered; indeed, the needle is no longer sterile once it passes through the working channel, potentially inoculating bacteria in the pleural space. However, the risk is low, and antibiotic prophylaxis may be used to prevent this complication.

In addition, in loculated pericardial effusion, several case reports also support the feasibility of EBUS-TBNA- or EUS-B-FNA-guided drainage [94,95,96,97].

Finally, even fluid below the diaphragm could be evaluated. Indeed, it is described as a single case report of ascites aspiration with EUS-B-FNA with no complications [98].

### 5.5. Thyroid Lesions

Intrathoracic goiter and thyroid lesions can represent a clinical challenge, as malignancy is often in the differential diagnosis. The percutaneous approach, which is applicable in the diagnostic workout of common thyroid lesions, is not suitable for intrathoracic lesions, which require mediastinoscopy or surgical biopsy. In these conditions, less invasive procedures, such as EBUS, might be considered [99]. In a case series by Madan and colleagues, aspiration of the thyroid lesion via EBUS-TBNA, which was not the primary indication for the procedure in more than half of the patients, was adequate in all the samples. Most of the procedures were safe except for one case complicated by an iatrogenic thyroid abscess [99]. This risk is greater for cystic lesions, and antibiotic prophylaxis is advisable [100].

### 5.6. Intratumoral Therapy

A range of oncogene-driven drugs is currently available in the therapeutic armamentarium of lung cancer. When molecular analysis does not identify a specific target, the immune status of NSCLC can be targeted. Standard chemotherapy protocols, with and without radiotherapy, still represent an optional or complementary approach. Yet, lung cancer remains the leading cause of death worldwide [101]. Among the possible strategies, intratumoral therapy has the potential to avoid checkpoint resistance and reduce systemic side effects [102]. Moreover, the injection permits to achieve a higher local concentration of the drug compared to that delivered systematically, up to 30-fold more [102,103], reducing side effects because of the decreased systemic concentration [104].

In the last decades, our knowledge of endobronchial chemotherapy delivery has grown substantially, widening the available approaches to malignant airway obstruction. 

Conventional TBNA has been used for several years for injection into the airways, lung parenchyma and surrounding tissues. Apart from the delivery of cancer therapies, transbronchial needles have been utilized for the injection of corticosteroids, antimicrobials, tranexamic acid, and radioisotopes to identify lung lesions or tissue sealants to treat bronchopleural fistula [102]. In recent years, the use of endobronchial ultrasound has optimized the technique of ultrasound-guided transbronchial needle injection (EBUS-TBNI) through the ability to reach not only endobronchial lesions but also those outside the airways or lymph nodes and the visualization of drug delivery, thus minimizing complications such as puncture of blood vessels or mediastinal structures [105].

More recently, this technique has been applied to deliver both standard chemotherapy [106,107,108] and immunotherapy [109]. This technique might be further improved by preoperative stratification of dose and site of injection according to computational modeling, which allows planning of the procedure according to the precise estimation of tumor size and localization [110].

EUS-B approach for the cisplatin injection in a patient with a critical, nearly complete tracheal occlusion and with atlantoaxial instability that contraindicates rigid bronchoscopy, has also been reported [111].

Uncommon applications of endoscopic ultrasound are summarized in Table 1. 

Figure 2 displays all sites that can be evaluated and sampled by the EBUS scope.

## 6. Novel Devices That Can Improve EBUS TBNA Performance and Future Perspectives

New complementary tools have been developed to obtain biopsy specimens for histological analysis for EBUS TBNA.

The Acquire^®^ 22 G fine needle (Boston Scientific Co., Natick, MA, USA) is characterized by a Franseen tip 22 G FNB device equipped with three cutting edges. In a prospective study, this needle has shown a better performance than FNA when applied to sample solid lesions by EUS [112]. Its feasibility and potential advantages have also been assessed in granulomatous diseases by Balwan and colleagues, who found a diagnostic yield of 95% [113]. An ongoing randomized trial is currently assessing whether, as compared to TBNA, the biopsies obtained by this needle provide larger or better quality tissue (NCT04200105).

The ProCore^®^ needle (Cook Medical, Bloomington, IN, USA) has a core trap designed to collect a histological sample by shearing material from the lesion during the needle motion [114]. This needle seems to allow similar diagnostic accuracy as a standard 22-gauge needle for both malignancy and suspected sarcoidosis [114,115].

An additional tool to increase the size of the tissue sampled is the EBUS-guided intranodal forceps biopsy (IFB), which is performed by introducing the mini forceps through the initial hole made by the TBNA needle. Compared to EBUS-TBNA, the addition of IFB provided a higher diagnostic yield, especially in patients with sarcoidosis (93% vs. 58%, *p* < 0.001, respectively) and lymphoma (86% vs. 30%, *p* = 0.03, respectively) [96]. The complication rates (pneumothorax and pneumomediastinum 1%, bleeding 0.8%) were higher than those reported with EBUS-TBNA alone (pneumothorax, 0.03% and bleeding, 0.68%) [116].

Of note, EBUS-IFB provides histology samples that, in parallel to those permanently fixed in formalin, can be instantly frozen, giving intraprocedural feedback [117]. Overall, IFB represents a complementary tool that might be offered to those patients in which TBNA could not provide the required amount of tissue to properly characterize tissue avoiding surgery.

Another novel tool to sample the lymph node is transbronchial cryobiopsy under EBUS guidance, wherein a 1.1 mm cryobiopsy probe is inserted in the hole created by the TBNA, with a 3 s freezing time [118]. 

The first randomized control trial (n = 197) was performed in 2021 by Zhang and colleagues, who compared standard EBUS-TBNA to cryobiopsy to sample mediastinal lesions of at least 1 cm [119]. The overall diagnostic yield was significantly higher with cryobiopsy (91.8% vs. 79.9%; *p* = 0.001). However, on subgroup analysis, when only metastatic lymph nodes were analyzed, the diagnostic yields of the two procedures were similar (94.1% vs. 95.6%, *p* = 0.58). Conversely, cryobiopsy was more accurate than EBUS-TBNA in benign disorders (80.9% vs. 53.2%, *p* = 0.004, respectively) and uncommon tumors (91.7% vs. 25.0%, *p* = 0.001, respectively). Notably, in the latter group, which was made mainly by lymphoma, mediastinal cryobiopsy was associated with a diagnostic rate of nearly 90% and complete subtyping [120].

Moreover, in cases of lung cancer, almost all cryobiopsy material (93.3%) was suitable for gene mutation PCR testing, compared to 73.5% of the material obtained by TBNA, which is consistent with previous reports [17,121].

Interestingly, no differences in diagnostic yield were found between the “Cryobiopsy first” and “TBNA first” groups. Two pneumothoraces and one pneumomediastinum were reported, highlighting the safety of the procedure. 

A recent randomized multicenter study (n = 297) assessed the overall diagnostic yield of the addition of cryobiopsy vs. standard needle sampling alone for mediastinal lesions [122]. The complementary use of EBUS cryobiopsy provided a 12% increase in the overall diagnostic yield compared to EBUS-TBNA alone. In a subgroup analysis, the combined approach provided a higher diagnostic yield in benign lesions but not in mediastinal metastasis, including uncommon tumors [122].

The combined approach also allowed more precise molecular and immunological analyses of lung cancer and a good safety profile. 

## 7. Conclusions

Since its first application in 2002, real-time transbronchial vision provided by EBUS has revolutionized the diagnostic approach to lung and mediastinal lesions. The development and implementation of novel complementary technologies, briefly reviewed here, have improved the diagnostic yield of the EBUS scope and expanded its possible applications to several diseases involving the mediastinum and beyond. The increasing need for tissue for diagnostic assessment, coupled with the growing knowledge of pathological and molecular features of lung cancer, will further push the optimization of the EBUS scope as a crucial diagnostic tool in a range of respiratory diseases.

## Figures and Tables

**Figure 1 diagnostics-13-02565-f001:**
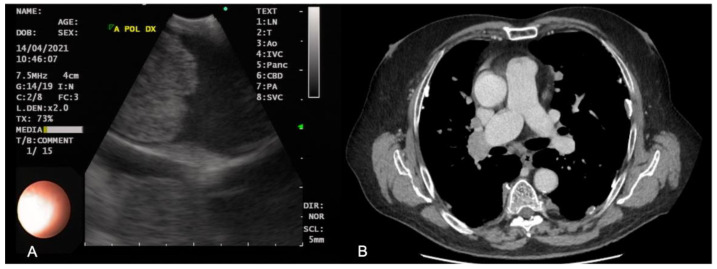
Echoendoscopic (**A**) and CT (**B**) images of pulmonary embolism of the right main pulmonary artery.

**Figure 2 diagnostics-13-02565-f002:**
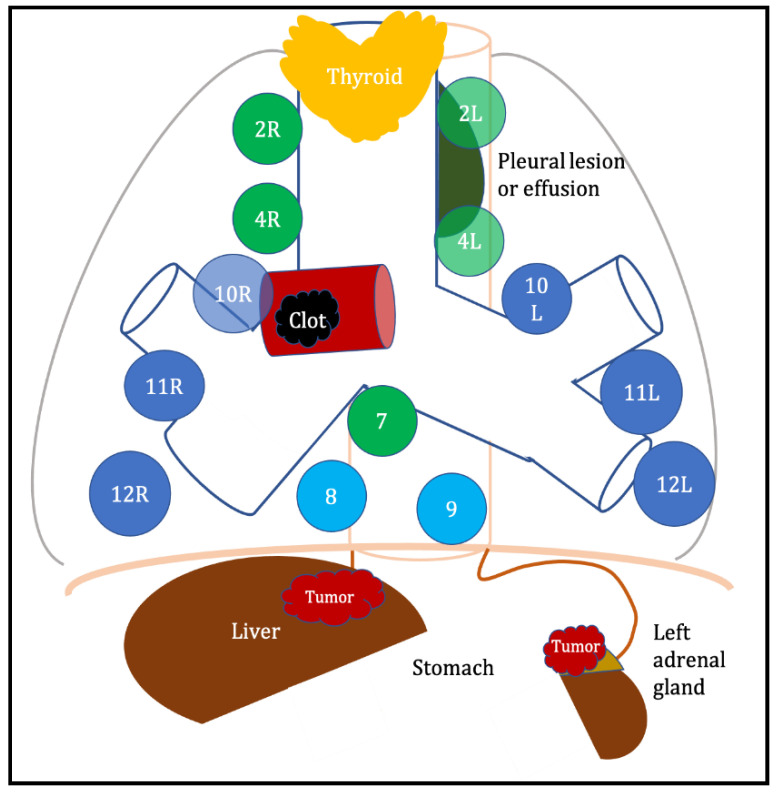
Sites potentially accessible by EBUS scope. Blue circles represent the main mediastinal stations accessible only by EBUS, light blue circles represent lymph node stations accessible only by EUS-B, and green circles represent lymph node stations accessible by both EBUS and EUS-B.

**Table 1 diagnostics-13-02565-t001:** Uncommon applications of endoscopic ultrasound.

Indication	Specific Complementary Tools	Advantages	Disadvantages	References
Mediastinal vessels evaluation	Color Doppler; echocardiography	Possibility to be performed bedside feasibility; not contraindicated in kidney failure, pregnancy and allergy to contrast medium. Diagnosis of neoplastic intravascular involvement	Poor sensibility in peripheral pulmonary embolism	[78,79,80,81,82]
Transvascular needle aspiration	Color Doppler	Reach lesion behind great vessels	Risk of bleeding	[83]
Pleural and cardiac lesions	-	Minimally invasive compared to surgery	Limited to lesions located next to airways or esophagus	[84,91]
Pleural/pericardial effusion and ascites	-	Valuable option in loculated effusions	Potential infection	[92,93,94,95,96,97,98]
Thyroid lesions	-	Valuable option in intrathoracic goiter/thyroid lesions	Potential infection	[99,100]
Intratumoral therapy	Computational modeling to design delivery strategy	Higher concentration at tumor site with lower systemic side effects	Risk of extravasation and airway irritation	[101,102,103,104,105,106,107,108,109,110,111]

## Data Availability

Figures and table are available from the corresponding authors upon reasonable request.

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
