# Peer review of "Clinical Applications of Endobronchial Ultrasound (EBUS) Scope: Challenges and Opportunities"

_diagnostics, 2023, doi:10.3390/diagnostics13152565_

Round 1

Reviewer 1 Report

The review has 2 general areas: The conventional use of EBUS in the diagnosis of mediastinal disease, and the new and unusual applications in sampling other sites such as the liver and adrenal.

The first component offers no new information, and previous reviews have covered this topic.  The second component is interesting, but needs a more detailed review and discussion.

I recommend the authors focus and elaborate on the second component in a revised manuscript.

Some English editing is needed, even thought this is not a major concern in the current manuscript.

Author Response

The review has 2 general areas: The conventional use of EBUS in the diagnosis of mediastinal disease, and the new and unusual applications in sampling other sites such as the liver and adrenal.

The first component offers no new information, and previous reviews have covered this topic.  The second component is interesting, but needs a more detailed review and discussion.

I recommend the authors focus and elaborate on the second component in a revised manuscript.

Following the reviewer’s suggestions, we amplified the second component of the manuscript (especially EUS-B-FNA and uncommon use of the EBUS scope) adding new details and new comments.

We also performed a revision of the English language to make the manuscript more readable.

Reviewer 2 Report

The manuscript tries to cover endoscopic biopsy techniques including cryobiopsy, sampling both via trachea and esophagus /EBUS-TBNA and EUS and EUS-B-FNA), diagnosis and staging of lung cancer, different clinical situations, molecular testing, elastography, ROSE, specimen preservation, intratumorally therapy plus many other topics on a few pages. I would have preferred to focus on a few of these – for example EBUS and EUS-B – and to leave out some of the other topics for example the historical aspects.

I suggest that the headline is changed to a headline that better covers the manuscript if the authors really insist on covering all these topics. Or maybe better to focus on a few.

Please go carefully through the text and differentiate clearly between EBUS-TBNA, EUS-B-FNA and EUS for every statement. For example, it is stated, that “the indications for EBUS have extended beyond the thoracic boundaries to sample lesions of the liver, left adrenal and retroperitoneal lymph nodes via the gastroesophageal tract”. This is not correct. In the esophagus the procedure is either EUS-FNA or EUS-B-FNA. In the airways it is EBUS-TBNA. EUS‑guided fine‑needle aspiration (EUS‑FNA) can be performed either with a conventional gastrointestinal echoendoscope (EUS) or by using the EBUS‑scope in the esophagus (EUS‑B). Avoid confusion and use the correct terms. Please go carefully through the manuscript and decide which of these techniques you talk about in every case.

Please rephrase the expressions “non-ordinary applications” and “unusual” applications. I do not think these expressions give sense. I do not know the definition of “ordinary” or “usual” applications. It must be possible to find a better way to say this.

 I suggest adding the below mentioned information to the manuscript:

The European Respiratory Society (ERS) has launched a structured evidence-based training program in EBUS‑TBNA based on training in simulators instead of patients ending with a validated test (Farr A, Clementsen PF, Herth F, Konge L, Rohde G, Dowsland S, Annema J. Endobronchial ultrasound: Launch of ERS structured training program. Breathe 2016; 12: 217 – 220). No educational program for EUS-B-FNA exists.

Bronchoscopy, EBUS-TBNA and EUS-B-FNA can be performed in the same session and with the patient under conscious sedation (Clementsen PF, Bodtger U, Konge L, Christiansen IS, Nessar R, Salih GN, et al. Diagnosis and staging of lung cancer with the use of one single echoendoscope in both the trachea and the esophagus: A practical guide. Endosc Ultrasound 10.4103/EUS-D-20-00139

Under Section 5 EUS-B Fine Needle Aspiration: With the use of EUS‑B‑FNA it is possible to biopsy the left adrenal gland (the references are already in the manuscript), lung tumors (Skovgaard Christiansen I, Kuijvenhoven JC, Bodtger U, et al. Endoscopic ultrasound with bronchoscope‑ guided fine needle aspiration for the diagnosis of paraesophageally located lung lesions. Respiration 2019; 97:277‑83), retroperitoneal lymph nodes, and the liver (Christiansen IS, Bodtger U, Naur TM, et al. EUS‑B‑FNA for diagnosing liver and celiac metastases in lung cancer patients. Respiration, 2019;98:428‑33). Also, case reports describe EUS-B-FNA from pleura (Bibi R, Bodtger U, Nessar R, et al. Endoscopic ultrasound‑guided pleural biopsy in the hands of the pulmonologist. Respirol Case Rep 2020;8:e00517), pancreas (https://doi.org/10.1016/j.rmcr.2023.101833) and to aspirate ascites (Nessar R, Toennesen LL, Bodtger U, et al. Endoscopic ultrasound‑guided ascites aspiration in the hands of the chest physician using the EBUS endoscope in the oesophagus. Respir Med Case Rep 2020; 29:100998) and pericardial fluid (Christiansen IS, Clementsen PF, Petersen JK, et al. Aspiration of pericardial effusion performed with EUS‑B‑FNA in suspected lung cancer. Respiration 2020; 99:686‑9).

delete superfluous words

Author Response

The manuscript tries to cover endoscopic biopsy techniques including cryobiopsy, sampling both via trachea and esophagus /EBUS-TBNA and EUS and EUS-B-FNA), diagnosis and staging of lung cancer, different clinical situations, molecular testing, elastography, ROSE, specimen preservation, intratumorally therapy plus many other topics on a few pages. I would have preferred to focus on a few of these – for example EBUS and EUS-B – and to leave out some of the other topics for example the historical aspects.

I suggest that the headline is changed to a headline that better covers the manuscript if the authors really insist on covering all these topics. Or maybe better to focus on a few.

According to the reviewers’ suggestion, we changed the title of the article and some headline in order to be more consistent with the topic covered.

Please go carefully through the text and differentiate clearly between EBUS-TBNA, EUS-B-FNA and EUS for every statement. For example, it is stated, that “the indications for EBUS have extended beyond the thoracic boundaries to sample lesions of the liver, left adrenal and retroperitoneal lymph nodes via the gastroesophageal tract”. This is not correct. In the esophagus the procedure is either EUS-FNA or EUS-B-FNA. In the airways it is EBUS-TBNA. EUS‑guided fine‑needle aspiration (EUS‑FNA) can be performed either with a conventional gastrointestinal echoendoscope (EUS) or by using the EBUS‑scope in the esophagus (EUS‑B). Avoid confusion and use the correct terms. Please go carefully through the manuscript and decide which of these techniques you talk about in every case.

We agree with the reviewer's comment and we modified all the incorrect use of the term “EBUS” in the title, in the abstract and in the text.

Please rephrase the expressions “non-ordinary applications” and “unusual” applications. I do not think these expressions give sense. I do not know the definition of “ordinary” or “usual” applications. It must be possible to find a better way to say this.

Again, we agree with the reviewer and, as suggested, we changed the expression “non-ordinary” and “unusual” to “uncommon”.

 I suggest adding the below mentioned information to the manuscript:

The European Respiratory Society (ERS) has launched a structured evidence-based training program in EBUS‑TBNA based on training in simulators instead of patients ending with a validated test (Farr A, Clementsen PF, Herth F, Konge L, Rohde G, Dowsland S, Annema J. Endobronchial ultrasound: Launch of ERS structured training program. Breathe 2016; 12: 217 – 220). No educational program for EUS-B-FNA exists.

Bronchoscopy, EBUS-TBNA and EUS-B-FNA can be performed in the same session and with the patient under conscious sedation (Clementsen PF, Bodtger U, Konge L, Christiansen IS, Nessar R, Salih GN, et al. Diagnosis and staging of lung cancer with the use of one single echoendoscope in both the trachea and the esophagus: A practical guide. Endosc Ultrasound 10.4103/EUS-D-20-00139

Under Section 5 EUS-B Fine Needle Aspiration: With the use of EUS‑B‑FNA it is possible to biopsy the left adrenal gland (the references are already in the manuscript), lung tumors (Skovgaard Christiansen I, Kuijvenhoven JC, Bodtger U, et al. Endoscopic ultrasound with bronchoscope‑ guided fine needle aspiration for the diagnosis of paraesophageally located lung lesions. Respiration 2019; 97:277‑83), retroperitoneal lymph nodes, and the liver (Christiansen IS, Bodtger U, Naur TM, et al. EUS‑B‑FNA for diagnosing liver and celiac metastases in lung cancer patients. Respiration, 2019;98:428‑33). Also, case reports describe EUS-B-FNA from pleura (Bibi R, Bodtger U, Nessar R, et al. Endoscopic ultrasound‑guided pleural biopsy in the hands of the pulmonologist. Respirol Case Rep 2020;8:e00517), pancreas (https://doi.org/10.1016/j.rmcr.2023.101833) and to aspirate ascites (Nessar R, Toennesen LL, Bodtger U, et al. Endoscopic ultrasound‑guided ascites aspiration in the hands of the chest physician using the EBUS endoscope in the oesophagus. Respir Med Case Rep 2020; 29:100998) and pericardial fluid (Christiansen IS, Clementsen PF, Petersen JK, et al. Aspiration of pericardial effusion performed with EUS‑B‑FNA in suspected lung cancer. Respiration 2020; 99:686‑9).

We thank the reviewer for these suggestions.

The first and the second paragraph were included in the manuscript with minimal changes according to the current text.

The third paragraph has been added through the manuscript to maintain the structure of the review. We hope that this is acceptable for the reviewer.

Regarding the comments on the English language, we tried to rewrite the entire manuscript, as appreciable in the text, reducing also the superfluous words.